# Proteomic Profiles of Whole Seeds, Hulls, and Dehulled Seeds of Two Industrial Hemp (*Cannabis sativa* L.) Cultivars

**DOI:** 10.3390/plants13010111

**Published:** 2023-12-30

**Authors:** Jan Bárta, Pavel Roudnický, Markéta Jarošová, Zbyněk Zdráhal, Adéla Stupková, Veronika Bártová, Zlatuše Krejčová, Jan Kyselka, Vladimír Filip, Václav Říha, František Lorenc, Jan Bedrníček, Pavel Smetana

**Affiliations:** 1Department of Plant Production, Faculty of Agriculture and Technology, University of South Bohemia, 370 05 České Budějovice, Czech Republic; barta@fzt.jcu.cz (J.B.); jarosovam@fzt.jcu.cz (M.J.); stupkova@fzt.jcu.cz (A.S.); 2Mendel Centre of Plant Genomics and Proteomics, Central European Institute of Technology, Masaryk University, 625 00 Brno, Czech Republic; pavel.roudnicky@ceitec.muni.cz (P.R.); zbynek.zdrahal@ceitec.muni.cz (Z.Z.); 3HEMP PRODUCTION CZ, Ltd., 262 72 Chraštice, Czech Republic; krejcova@ekonopi.cz (Z.K.); riha@ekonopi.cz (V.Ř.); 4Department of Dairy, Fat and Cosmetics, University of Chemistry and Technology, 166 28 Prague, Czech Republicvladimir.filip@vscht.cz (V.F.); 5Department of Food Biotechnology and Agricultural Products Quality, Faculty of Agriculture and Technology, University of South Bohemia, 370 05 České Budějovice, Czech Republicsmetana@fzt.jcu.cz (P.S.)

**Keywords:** industrial hemp, hulls, dehulled seeds, proteomic profile, 11S globulin

## Abstract

As a source of nutritionally important components, hemp seeds are often dehulled for consumption and food applications by removing the hard hulls, which increases their nutritional value. The hulls thus become waste, although they may contain valuable protein items, about which there is a lack of information. The present work is therefore aimed at evaluating the proteome of hemp (*Cannabis sativa* L.) at the whole-seed, dehulled seed, and hull levels. The evaluation was performed on two cultivars, Santhica 27 and Uso-31, using LC-MS/MS analysis. In total, 2833 protein groups (PGs) were identified, and their relative abundances were determined. A set of 88 PGs whose abundance exceeded 1000 ppm (MP88 set) was considered for further evaluation. The PGs of the MP88 set were divided into ten protein classes. Seed storage proteins were found to be the most abundant protein class: the averages of the cultivars were 65.5%, 71.3%, and 57.5% for whole seeds, dehulled seeds, and hulls, respectively. In particular, 11S globulins representing edestin (three PGs) were found, followed by 7S vicilin-like proteins (four PGs) and 2S albumins (two PGs). The storage 11S globulins in Santhica 27 and Uso-31 were found to have a higher relative abundance in the dehulled seed proteome (summing to 58.6 and 63.2%) than in the hull proteome (50.5 and 54%), respectively. The second most abundant class of proteins was oleosins, which are part of oil-body membranes. PGs belonging to metabolic proteins (e.g., energy metabolism, nucleic acid metabolism, and protein synthesis) and proteins related to the defence and stress responses were more abundant in the hulls than in the dehulled seeds. The hulls can, therefore, be an essential source of proteins, especially for medical and biotechnological applications. Proteomic analysis has proven to be a valuable tool for studying differences in the relative abundance of proteins between dehulled hemp seeds and their hulls among different cultivars.

## 1. Introduction

Hemp (*Cannabis sativa* L.), also known as industrial hemp, has long been cultivated as a source of solid and durable fibre in the stems, oily seeds, and active compounds useful in medicine [1,2]. Compared to *Cannabis indica*, most industrial hemp cultivars contain low levels of the drug tetrahydrocannabinol (THC), which has enabled an increase in its acreage in much of the world today. In addition to the valuable fibre, the seeds represent another important and economically exploitable component [3].

Hemp seeds are nutritionally very valuable; on average, whole seeds contain 25–35% fat, 20–28% protein, 25–37% carbohydrate (including fibre), 5–6% ash, and 4–8% moisture [1,3,4]. A large proportion of hemp seeds are used (mainly through cold pressing) to produce hemp oil, which is valued for its high content of polyunsaturated fatty acids. A number of papers [5,6,7,8,9] report a polyunsaturated fatty acid content of over 70% in the oil, mainly the essential n-6 linoleic acid (around 55%) and n-3 alpha-linolenic acid (12–18%) and minor levels of n-6 gamma-linolenic acid (1–3%) and n-3 stearidonic acid (<1%).

In addition to the oil, hemp seeds also contain easily digestible protein with a good amino acid composition, in which arginine plays an important role. Compared to soybean proteins, hemp seed proteins do not contain protease inhibitors in large amounts and are also considered less allergenic than proteins from other plant seeds [10]. Like the seeds of other dicotyledonous plants, hemp seed proteins are mainly represented by the globulin fraction (around 75%), with the rest being albumins [11,12].

Globulins are predominantly represented by the legumin-like 11S protein family [2,13], which, in hemp seed, is represented by edestin, which has a hexameric structure in its native state [12,14,15]. The edestin monomer has a molecular weight (MW) of 52–54 kDa and is composed of acidic (MW around 34 kDa) and basic (MW 20–25 kDa) polypeptides, which are linked together by a disulphide bond [12]. Edestin is a storage protein in the seed and is present in 60–80% of the total protein pool of hemp seeds [2,14]. The minor fraction of globulins is represented by the trimeric 7S protein (MW monomer 47-48 kDa), one of the vicilin-like proteins [4,16]. Albumin is mainly represented by a 10 kDa protein (2S albumin), which is 18% by weight of sulphur amino acids (cysteine, methionine) and, therefore, belongs to the group of sulphur-rich proteins [12].

Regarding the study of the protein profile of hemp seeds, several proteomic studies are available that deal with the analysis of proteins in whole seeds [17,18], in protein isolates [4,10], or in products after extrusion treatment [19]. Some studies [20,21,22] also present proteomic techniques as a tool suitable for verifying the authenticity of foods and their composition or assessing foods for the presence of allergens, which can be used to verify the addition of flours prepared from oilseeds (including hemp seeds) to food products. Most of the works [4,10,18,19,22] mentioned edestins or 11S globulins as major proteins in hemp seeds. Other proteins identified included proteins responsible for the biocatalysis of metabolic pathways, proteins related to nucleic acid and protein synthesis, proteins associated with membrane structures and functions, and stress proteins [17,18].

The above-mentioned studies [4,10,17,18,19] addressed protein number and identification using available databases but did not compare the relative abundance of identified items within the hemp seed protein pool. Moreover, in these studies, only one hemp cultivar was always used as the source of seed material, so information on varietal differences is lacking. Hemp seeds (more correctly, hemp achene) are dehulled for direct consumption and other food applications, which involves removing the hard, difficult-to-eat hull consisting of the pericarp and the seed coat layers, which contain primarily insoluble and indigestible fibre. On the other hand, dehulling increases the concentration of fat and protein in the dehulled seed [23]. It may thus alter the relative abundance of protein items in the dehulled seed compared to the whole (unhulled) seed. The hulls themselves can be finely ground, and the resulting powder (flour) can be used to fortify bakery, meat, or other food products with fibre and vegetable protein. However, the protein profiles of the hulls in terms of the presence and relative abundance of the different protein items have not been investigated, and information is completely lacking. Knowledge of the differences between the proteomes of the pericarp and the dehulled seed may be essential from the biological, nutritional, and technological points of view, as well as from the point of view of potential health risks and food safety.

Nowadays, the gel-free approach utilising LC-MS/MS analysis is the most efficient tool allowing the deep coverage of proteomes. The perspectives and potential of seed proteomics were recently reviewed by Smolikova et al. [24].

To increase the knowledge about the possibilities of using hemp seeds, dehulled seeds, and hulls as protein sources, the objectives of this study were (i) to perform proteomic characterisation at the level of whole seeds, dehulled seeds, and hulls; (ii) to evaluate the differences in the relative abundance of significant proteins and protein classes in these seed products in two industrial hemp cultivars.

## 2. Results and Discussion

### 2.1. Protein and Fat Contents in Original and Defatted Seed Product Samples

Protein and fat are the most valuable components of hemp seeds. As shown in Table 1, for both cultivars evaluated, Uso-31 and Santhica 27, the protein content of whole original seeds was found to be around 26–28% of dry matter (DM), and the fat content was around 29% of DM. The fat content of the whole seed corresponds to the usually reported data [1]; the protein content found is higher than or at the upper limit of the already-reported values of 20–25% [1] or 21.3–28.1% [3].

The dehulled original (non-defatted) seed has increased protein and fat contents, with values of 35% and 49%, respectively, in the two cultivars evaluated compared to the original whole seed (Table 1), which is roughly consistent with the literature reports [23,25]. In contrast, according to the literature data, significantly lower protein and fat contents were found in hulls, which have a high fibre content [25]. In the case of hulls, the effect of the cultivar on the separation of the hull from the rest of the seed seems to be more pronounced, as the reduction in the protein and fat contents of hulls relative to the whole seed does not occur in the same proportion in the two cultivars evaluated. In the case of the cultivar Uso-31, there was a reduction in the protein content of the hulls from 25.8 to 13.4% of DM (relative reduction to 51.9% content of original seed) and a reduction in the fat content from 29.4 to 6.2% (relative reduction to 21.1%). In the case of Santhica 27, there was a lower reduction in the protein and fat contents of the hulls, close to the values in the whole seed: the protein content was reduced from 27.7 to 19.2% (relative reduction to 69.3%), and the fat content was reduced from 28.7 to 15.4% (relative reduction to 53.7%).

For the defatted variants of the samples, which were subsequently subjected to detailed proteomic analysis, the fat content was reduced to below 1% of dry matter in almost all samples due to defatting with an organic solvent. The protein content increased in all samples due to defatting, with the dehulled seed samples showing an increase in protein content to above 65%, the value reported as the minimum protein content for protein concentrate products [26]. The increase in protein content is correlated with the original fat content removal. Thus, the highest growth in protein content was obtained in the dehulled seeds, from 35.4 to 71.7% of DM for the cultivar Uso-31 and from 35.6 to 68.5% of DM for the cultivar Santhica 27. Shen et al. [27] also observed increased protein content in flour and protein isolates prepared from dehulled seeds compared to the non-dehulled variants. However, they found only 41.8% protein content (Nx6.25) and 8.8% residual fat content (at 4.3% moisture content) for flour prepared from dehulled seeds after mechanical defatting, which is lower than our findings.

The results indicate that the mechanical separation of the hulls from hemp seeds is one of the two processes that significantly affect the production of hemp seed products with different primary nutrient contents. The second process should be defatting [28], either through mechanical pressing and fat extraction using an organic solvent or direct solvent use. Combining both processes (dehulling and defatting) can be essential to obtain concentrates with a higher protein content.

### 2.2. Characterisation of Seed Protein Pool

The essential features of the protein profiles of defatted seed, defatted hull, and dehulled seed samples are evident after one-dimensional SDS-PAGE (Figure 1). Four main zones of protein bands (A–D) are clear for all three types of samples of the two cultivars evaluated. However, the protein bands in the four zones mentioned above have different intensities. The A-zone region, with the weakest staining intensity, should represent a 7S vicilin-like protein with an MW of 47–48 kDa, similarly reported by several studies [12,29,30,31]. The most intense protein bands are those in zones B and C, which represent acidic (MW 30–35 kDa) and basic (MW 18–20 kDa) polypeptides of the 11S protein (edestin) monomer. The D zone proteins (MW 15 kDa and less) should represent the albumin fraction of the protein. The presence of edestin and the albumin fraction in SDS-PAGE gels has also been described by Liu et al. [29], Liu et al. [30], and Fang et al. [32].

Based on the evaluation of the proteomic data, 2833 protein groups (all belonging to *Cannabis sativa* L.) were identified, and their relative protein abundances in individual samples were determined and expressed in ppm. For further assessment of the protein pool, only protein groups (hereafter referred to as proteins or protein items) with a relative abundance greater than 1000 ppm (0.1%) in at least one of the three independent replicates for one of the three sample types (original or defatted whole seed, hulled seed, or hull) were considered. Eighty-eight of these major proteins were found (hereafter referred to as collection MP88). Their list, names, and primary data (molecular weight; the name of the corresponding protein family), including a more detailed characterisation of the proteins in the areas of biological process, molecular function, and cellular component, obtained from the UniProt KB database, are given in Table 2. Based on this detailed characterisation, the MP88 proteins were divided into ten protein classes: seed storage proteins (SP), oleosins (OL), other membrane components (MC), proteins involved in energy and metabolism (EM), translation-related proteins (TR), proteins related to DNA and RNA metabolism (DM), stress response and defence proteins (SD), proteins related to photosynthesis (PS), cytoskeleton and transport proteins (CT), and uncharacterised proteins (UP).

It should be noted that the number of proteins classified in these classes and their relative abundances in the samples of the three types reflect the condition of the maturing/mature seed. This state is associated with physiological processes involving the accumulation of di- and oligosaccharides, the synthesis of storage proteins, LEA proteins and heat-shock proteins, and the activation of antioxidant defences, in addition to the increasing dry matter content and the physical, structural changes in the cells [33]. In comparison to our experimental setup, Park et al. [17], in their study of the proteomic profiling of Cheungsam (*Cannabis sativa* L.) seeds using a combination of 2D PAGE and nano-LC-MS/MS, found 168 identified unique protein spots out of a total of 1102 spots resolved by the 2D PAGE separation of the extracted hemp seed protein mixture due to the lack of hemp protein sequences. Park et al. [17] used the database information of rice and Arabidopsis genomes, which had been completely sequenced at the time, to identify the proteins. The identified proteins were classified into 13 categories according to their function, as listed in the SWISS-PROT and NCBI databases.

#### 2.2.1. Seed Storage Proteins

The SP class represents the most abundant group: seed storage proteins. The MP88 dataset included nine proteins. Three proteins (A0A7J6GWL5, A0A7J6DTA7, A0A7J6E205) were classified into the 11S family of seed storage proteins, commonly referred to as edestin in hemp seeds [12,14]; then, two proteins (A0A7J6H292, A0A7J6DXD1) were assigned to 2S albumins, and the remaining four proteins (A0A7J6H2R3, A0A7J6GLH5, A0A7J6HAT3, and A0A7J6G321) belong to the globulin family, and some of them belong to the 7S family (see Table 2). The names of these proteins do not directly imply that they are 7S proteins, but additional information found in the UniProt KB database suggests their similarity to 7S vicilin-like proteins. The occurrence of the three types of seed storage proteins mentioned above agrees with most papers that have investigated hemp seed proteins [10,16,22,34].

The main storage protein in hemp seeds is edestin. Docimo et al. [34], in their study on the molecular characterisation of the edestin gene family, found seven genes encoding seven forms of edestin, which can be divided into two types of edestin based on the sequence similarity—edestin 1 (includes four forms) and edestin 2 (involves three forms). In subsequent work [16], edestin type 3 was additionally found, which resembles type 1 more than type 2. The three above items of 11S proteins that we found are not directly identified as edestin in the UniProt KB database version used for database searching (they are named Cupin type-1 domain-containing protein) but show high sequence similarity to the edestins described in previous works [16,34]. The storage protein A0A7J6GWL5 has, according to the UniProt KB database, an MW of 109.15 kDa and is composed of a sequence of 953 amino acids, showing a close identity (99.8%) to edestin 1 (A0A090DLH8) in region 8-511, as reported in the UniProt database by, e.g., Mamone et al. [10] and Kotecka-Majchrzak et al. [22]. The second found 11S storage protein item (A0A7J6DTA7) has an MW of 52.03 kDa, is composed of 456 amino acids, and is consistent with forms of edestin 2—with item A0A090DLI7 at 98.1%, A0A090CXP9 at 97.8%, and A0A090CXP8 at 98.1%. The form of edestin 3 was not found in our experiment.

#### 2.2.2. Proteins Involved in Energy and Metabolism

The class of proteins involved in energy and metabolism is represented by 15 proteins in the MP88 set. The most significantly represented items are those related to glycolysis, the tricarboxylic acid cycle (TCA), and other central metabolic processes, which are closely related to the physiological state of mature stored seed with preserved germination potential [33]. The importance of enzymes related to glycolysis, TCA, and the glyoxylate cycle for the metabolism of maturing oilseeds was also confirmed by Hajduch et al. [35].

Specifically, of the glycolytic enzymes, two glyceraldehyde-3-phosphate dehydrogenase items were represented in the MP88 set (items A0A7J6G6Z3 and A0A7J6HK40), in addition to triosephosphate isomerase (A0A7J6E4U9), fructose-bisphosphate aldolase (A0A7J6E5J2), and phosphopyruvate hydratase or enolase (A0A7J6GRW8). The TCA cycle is represented by two items of malate dehydrogenase (A0A7J6EZ77 and A0A7J6E8J3), and the glyoxylate cycle is represented by the enzyme malate synthase (A0A7J6FNP6).

Other important items with catalytic activity are involved in protein modification (protein disulfide-isomerase (A0A7J6EJG0) and peptidyl-prolyl cis-trans isomerase (A0A7J6EG53)), nucleotide metabolism (nucleoside-diphosphate kinase (A0A7J6HKH5)), etc. Other enzymes are related to electron and proton transfer in energy metabolism, e.g., NADH-cytochrome b5 reductase (A0A7J6HQA0) and ATP synthase (A0A0M5M1Z3). Due to their membrane localisation, these enzymes could also be classified as membrane components. Similarly, Park et al. [17] reported that the groups of proteins and enzymes related to basic metabolism and energy production in hemp seeds are the most numerous and diverse group of proteins. Their dataset also mainly lists enzymes of glycolysis, enzymes of the TCA cycle, and other important metabolic pathways.

#### 2.2.3. Membrane Proteins

The representation of membrane proteins, which include two classes, namely, oleosins (OL) and other membrane components (MC), is significant. The OL are proteins involved in the structure of oil-body membranes called oleosomes [36]. In hemp seed cells, these are droplet-like structures (droplets) with a diameter of 3–5 µm, in whose membranes the membrane-specific oleosin proteins play an important structural role, with an estimated MW of ≈15 kDa [37]. In addition to oleosins, caleosins and steroleosins have also been reported as fat-body membrane proteins [38]. In the group of major proteins, three oleosin items (A0A7J6EJ89, A0A7J6F0Y4, A0A7J6H4F6) were found with MWs ranging from 15.41 to 17.36 kDa, which together represent about 8% of hemp seed proteins. The second highest relative abundance of oleosins in the hemp seed protein pool (after seed storage proteins) is thus closely related to the high fat accumulation in hemp seed tissue. Among the items of the MC class, which is represented by eight proteins, one can find a caleosin protein with peroxygenase activity (A0A7J6F280) and, e.g., aquaporin (A0A7J6GSC3), which functions as a transmembrane channel. Most of the other proteins are designated as uncharacterised proteins.

#### 2.2.4. Proteins Involved in Stress Response and Defence

A total of 14 items were found in the MP88 dataset that can be classified as SD. These include LEA (late embryogenesis abundant) proteins (A0A7J6FL33, A0A7J6I6U6), including dehydrins (A0A7J6FXL4) as well as heat-shock proteins (A0A7J6GTA4, A0A7J6G382) and proteins associated with antioxidant protection against ROS (Reactive Oxygen Species), such as catalase (A0A7J6F3P9), peroxiredoxin (A0A7J6DT98), dehydroascorbate reductase (A0A7J6GJC9), glutaredoxin domain-containing protein (A0A7J6HTR6), and lactoylglutathione lyase or glyoxylase I (A0A7J6FPH5).

From the spectrum of stress-related proteins listed above, it is evident that the maturing/ripe seed and the developing/developed embryo must cope with progressive desiccation, temperature changes, and oxidative stress or oxygen deficiency. The occurrence of alcohol dehydrogenase in plants has been linked to oxygen deficiency.

Regarding seed defence, the occurrence of the enzyme rRNA N-glycosidase (A0A7J6DVP5) is significant. This enzyme generally belongs to the group of ribosome-inactivating proteins (RIPs), which are toxic because they inhibit proteosynthesis and are among the defence proteins [39]. Despite its toxicity, this group of proteins has the potential for possible applications in medicine or agriculture [40].

Park et al. [17] found, in their work on mature industrial hemp seeds, a similar spectrum of stress- or defence-related proteins—heat-shock proteins, glyoxylate reductase, dehydroascorbate reductase, alcohol dehydrogenase, and peroxiredoxin; in addition, they reported superoxide dismutase, which was also found in our samples, not in the MP88 set but in the set of all proteins. Aiello et al. [18] reported only Glyoxalase I and several variants of heat-shock proteins in their work focused on the proteomic characterisation of hemp seeds.

#### 2.2.5. Other Protein Classes

Common cellular proteins include those involved in nucleic acid metabolism and subsequent protein synthesis during translation. The items included in the DM class are mainly represented by histone protein subunits (A0A7J6E9G4, A0A7J6E6Z7, A0A7J6E9K3). The TR class has the most significant number of items in the set of major proteins, as well as the EM class; in most cases (20 out of 24 items), these are ribosome subunits, e.g., two items of 60S ribosomal protein (A0A7J6GTP8, A0A7J6E9P9), as well as 40S (A0A7J6GWW0) and 50S (A0A7J6H424) ribosomal proteins and several other proteins involved in translation, e.g., elongation factor 1-alpha (A0A7J6HCW3) or EF1_GNE domain-containing protein (A0A7J6HK4).

The chloroplastic chlorophyll a-b-binding protein represents the PS class, and the CT class is represented by a protein related to intracellular sterol transport and by a protein from the actin family. Six items listed in the database had no characteristics and were therefore assigned to the UP class.

### 2.3. Effect of Cultivar and Dehulling on the Hemp Seed Protein Profile

The relative abundance of protein items in the MP88 set in the total protein profiles of whole seeds, hulls, and dehulled seeds with respect to the two cultivars studied (Uso-31, Santhica 27) is shown in Table 3. The percentages of the protein classes (including the group of proteins not included in the MP88 set) are expressed using pie charts in Figure 2.

The presented results (Table 3) confirmed the assumption that the hemp seed’s main class of proteins is SP, especially the 11S globulin edestin. The relative abundance of 11S globulins (summed over the three MP88 items) in whole seeds was 58.6 and 63.2% for the Santhica 27 and Uso-31 cultivars, respectively, when expressed as percentages, approximately corresponding to the literature-reported range of 60–80% [2,14]. Liu et al. [29] reported a range of relative edestin proportions of 57–76% when studying nine protein isolates prepared from seeds of different *Cannabis sativa* cultivars/genotypes, but this is the relative representation of edestin among the storage proteins (11S, 7S, and 2S), not the representation of edestin in the overall protein profile of hemp seeds. All three 11S protein items (A0A7J6GWL5, A0A7J6DTA7, A0A7J6E205) of the MP88 set were found to have a conclusively (except in one case) higher relative abundance in hulled seeds than in hulls. Considering the accumulation of reserve compounds in the inner seed part, this finding confirms the reported role of 11S proteins and edestins as seed reserve proteins [2,14]. The finding is also interesting concerning the potential food applications of hemp seed proteins, as edestin is their most valuable component [34].

The protein items A0A7J6H2R3, A0A7J6GLH5, A0A7J6HAT3, and A0A7J6G321, which could be classified as 7S proteins or vicilin-like proteins, represent about 3% of the seed protein pool in whole seeds of both cultivars, which is significantly less than that reported by other authors: Sun et al. [4] reported 5% and Liu et al. [29] reported a range of 9–19% (but here, as mentioned above, it is the proportion of storage proteins). In contrast to the 11S proteins, the relative abundance of 7S proteins was inconclusively higher in the hulls than in the dehulled seeds in all cases (see Table 3).

The third group of storage proteins is represented by the 2S proteins (A0A7J6H292, A0A7J6DXD1), which are classified as albumins. Collectively, their representation in the whole seed was only 2.04% (cv. Uso-31) or 1.61% (cv. Santhica 27), which is lower than the values published by other authors: Sun et al. [4] reported 13%, and Liu et al. [29] reported a range of 12–24% within a set of nine cultivars (here again, the values are proportions of storage proteins). The predominance of 2S albumins in the hulls or dehulled seeds is not at all clear.

The second most represented group of SP is the class of oleosins. Because the inner part of the seed coat accumulates more oil as a natural energy reserve, one would expect a higher representation of oleosins in dehulled seeds as key proteins that are part of the fat bodies, in which fat is stored in the cells. The results showed that the total oleosins increased with dehulling, from 8.2 (WS) to 9.4% (DS) and from 7.5 (WS) to 8.9 (DS) in Uso-31 and Santhica 27, respectively. However, this is due to the presence of the oleosin A0A7J6EJ89, which is significantly higher in the dehulled seeds than in the hulls of both cultivars. The other two proteins (A0A7J6F0Y4, A0A7J6H4F6) have no clear differences in the different parts of the seed.

Either way, the oleosin protein fraction is important in food applications as a natural emulsification system [36,37,38] or as a lipid-based delivery system in the food industry or in cosmetics and pharmaceuticals [39,40].

The protein classes EM, SD, DM, TR, and CT are collectively more abundant in the hulls (proteins of some classes are even significantly more abundant) than in the hulled seeds, which generally indicates that higher metabolic activity is taking place in the hull cells. This can be explained by the function of the hull tissues, i.e., the pericarp and seed coat layers, which form the natural barrier of the hemp achene and are thus forced to react more strongly to environmental stimuli, particularly abiotic stressors. Most of the proteins classified as SD in the MP88 dataset have a higher relative abundance in the hull than in the dehulled seed, except for dehydrin (A0A7J6FXL4), glutaredoxin domain-containing protein (A0A7J6HTR6), and dehydroascorbate reductase (A0A7J6GJC9). N-glycosidase rRNA (A0A7J6DVP5) is significantly more abundant in hulls (8-fold in cv. Santhica 27, 30-fold in cv. Uso-31) than in hulled seeds. This finding confirms the function of tRNA N-glycosidase as a defence protein [39]. The higher abundance in the hull predisposes the hull to be a potential source of this enzyme for biomedical applications. The RIP proteins have been found to have important biological properties, such as anticancer, antiviral, and neurotoxic activities [40,41]. Both chlorophyll-binding proteins (A0A7J6GUI2, A0A7J6GSM8) were significantly overrepresented in the hulls (compared to dehulled seeds) of both cultivars, which can be explained by the fact that chlorophyll is mainly found in the innermost layer of the seed coat, which is part of the hull [42].

Dehulling is an important technological process that removes the hard hull of the hemp seed, allowing the seeds to be consumed directly and used more widely in food products. Removing the hull eliminates or reduces the contents of antinutrients, improves the sensory properties of the dehulled seeds, and enhances their digestibility [42,43]. Previous work [27,43,44], and our presented results, have confirmed that dehulling can also be an important technological step in producing hemp seed protein concentrates and isolates, while this step removes big amounts of ballast substances (especially fibre components). At the same time, the proteomic results indicate the importance of hemp seed hulls as a source of a specific protein. The hulls often represent only the waste from hemp seed processing, while it could be a source of a large spectrum of proteins or as a raw material for a fibre flour product. On the other hand, the finding of a higher abundance of N-glycosidase rRNA also points to the possible risks of using products from the hulls for (direct) human consumption or the fact that the cultivar itself and its selection may play a major role in the occurrence of risk proteins.

## 3. Materials and Methods

### 3.1. Hemp Seed Samples and Their Preparation

The seeds of two cultivars of industrial hemp (*Cannabis sativa* L.) with low THC content (up to 0.2%), Uso-31 (registered in EUPVP—Common Catalogue; National Listing Netherlands; variety ID: 214015) and Santhica 27 (registered in EUPVP—Common Catalogue; National Listing France; variety ID: 213998), from the 2020 harvest were obtained from Hemp Production CZ, Ltd. Three variants of samples from both cultivars were provided for analysis: whole hemp seeds, seed hulls and dehulled hemp seeds. Hemp seeds were dehulled using dehulling and separating equipment (TFYM 1000; Liaoning Qiaopai Machineries Co., Ltd., Jinzhou, China). Before analyses, the samples were disintegrated using a Grindomix GD200 knife mill (Retsch, Haan, Germany) at 10,000 rpm for 1 min. The ground samples were then subjected to defatting using the organic solvent n-hexane. To the sample powder in the plastic tube, n-hexane was added at a ratio of 1:3 (*w*/*v*), and after mixing and extraction for 2 h at room temperature (with shaking of the sample mixture every 30 min), centrifugation (rpm 4500, 10 min, 20 °C) was performed, the solvent was carefully removed from the tube, and then the whole fat extraction process was repeated twice. After defatting, the sample pellets were left free to dry in the laboratory fume hood. Three defatted sample variants—defatted whole seeds (WS), defatted hulls (H), and defatted dehulled seeds (DS)—were obtained by the above-described treatment.

### 3.2. Dry Matter, Protein, and Fat Contents

Dry matter, protein, and fat contents were determined for the original (with fat) and defatted versions of the hulls, whole seeds, and dehulled seeds. The dry matter content was determined gravimetrically by drying the samples at 105 °C for 3 h in an oven to constant weight. The protein content was determined using the modified Dumas combustion method using a rapid N cube (N/Protein analysis) instrument (Elementar Analysen System, Langenselbold, Germany). Each sample was analysed in triplicate, and the protein content was calculated as the nitrogen content multiplied by a factor of 6.25. The fat content was measured using the Soxhlet extraction method using an ANKOM XT 10 Extractor (ANKOM Technology, Macedon, USA), according to the manufacturer’s manual. Petroleum ether was used as an extraction reagent. The fat content was calculated from the weight differences in the sample before and after extraction.

### 3.3. Sodium Dodecyl Sulphate Polyacrylamide Gel Electrophoresis

The defatted sample variants were extracted with SDS extraction buffer (0.065 M Tris-HCl, pH 6.8; 2% (*w*/*v*) SDS; 5% (*v*/*v*) 2-sulphanylethanol) in a ratio of 1:10 (*w*/*v*) at 4 °C for 4 h. Protein separation was carried out in triplicate using cooled dual vertical slab units (SE 600; Hoefer Scientific Instruments, Holliston, MA, USA) with a discontinuous gel system (4% stacking and 12% resolving gel) in reducing conditions [45]. Protein detection was performed by using Coomassie Brilliant Blue R-250.

### 3.4. LC-MS/MS Analysis

Proteins for LC-MS/MS analysis were extracted in SDT buffer (4% SDS, 0.1M DTT, 0.1M Tris/HCl, pH 7.6) in a thermomixer (Eppendorf ThermoMixer C, 60 min, 95 °C, 750 rpm). After that, all samples were centrifuged (15 min, 20,000× *g*), and the supernatants (ca. 100 μg of total protein) were used for filter-aided sample preparation (FASP), as described elsewhere [46], using 0.75 μg of trypsin (sequencing grade; Promega). Proteins were digested overnight (18 h) at 37 °C. The resulting peptides were analysed using LC-MS/MS.

LC-MS/MS analyses of all peptides were performed using an UltiMate 3000 RSLCnano system connected to an Orbitrap Exploris 480 spectrometer (Thermo Fisher Scientific, Waltham, MA, USA). Prior to LC separation, tryptic digests were concentrated and desalted online using a trapping column (Acclaim PepMap 100 C18, 300 μm ID, 5 mm long, 5 μm particles, Thermo Fisher Scientific). After washing the trapping column with 0.1% FA, the peptides were eluted in backflush mode (flow 500 nL·min^−1^) from the trapping column onto an analytical column (EASY-Spray column, 75 μm ID, 250 mm long, 2 μm particles, Thermo Fisher Scientific), where peptides were separated using a 90 min gradient program (flow rate 300 nL·min^−1^, 3–37% of mobile phase B; mobile phase A: 0.1% FA in water; mobile phase B: 0.1% FA in 80% ACN). Both columns were heated to 40 °C.

MS data were acquired in a data-dependent strategy (cycle time 2 s). The survey scan range was set to *m*/*z* 350–2000 with a resolution of 120,000 (at *m*/*z* 200), a normalised target value of 250%, and a maximum injection time of 500 ms. HCD MS/MS spectra (isolation window *m*/*z* 1.2, 30% relative fragmentation energy) were acquired from *m*/*z* 120 with a relative target value of 50% (intensity threshold 5 × 103), a resolution of 15,000 (at *m*/*z* 200), and a maximum injection time of 50 ms. Dynamic exclusion was enabled for 45 s.

### 3.5. Proteomic Data Processing

For data processing, we used MaxQuant software (v2.0.3.0) [47] with an inbuilt Andromeda search engine [48]. A search was performed against protein databases of *Cannabis sativa* (30,194 protein sequences, version from 24 February 2022, downloaded from https://www.uniprot.org/proteomes/UP000583929, accessed on 1 January 2020) and cRAP contaminants (112 sequences, version from 22 November 2018, downloaded from http://www.thegpm.org/crap). Modifications were set for the database search as follows: oxidation (M), deamidation (N, Q), and acetylation (Protein N-term) as variable modifications, with carbamidomethylation (C) as a fixed modification. Enzyme specificity was tryptic with two permissible missed cleavages. Only peptides and proteins with a false discovery rate threshold under 0.01 were considered. Relative protein abundance was assessed using protein intensities calculated using MaxQuant. The intensities of reported proteins were further evaluated using a software container environment (https://github.com/OmicsWorkflows/KNIME_docker_vnc; version 4.1.3a). The processing workflow is available upon request, and it covers, in short, reverse hits and contaminant protein group (cRAP) removal, protein group intensities’ log2 transformation, and normalisation (loessF). For the purpose of this article, protein groups reported by MaxQuant are referred to as proteins or protein items.

### 3.6. Statistical Analysis

The program Statistica 12 (StatSoft Power Solutions Inc., Palo Alto, CA, USA) was used for the data analysis. Data were subjected to analyses of variance using the two-way ANOVA method, and the means were compared using the Tukey HSD test. Differences between the variants were considered significant at *p* < 0.05 unless stated otherwise.

## 4. Conclusions

In total, 2833 proteins were identified in this proteomic study of the whole seeds, hulls, and dehulled seeds of two industrial hemp (*Cannabis sativa* L.) cultivars. Of this number, only 88 proteins accounted for 81.5–91.4% of the relative quantity of total proteins. The proteins within this set, reflecting the physiological state of the mature and stored seeds, were classified into 10 classes according to molecular and biological functions. According to the literature, we confirmed most of all three types of storage proteins—11S (edestin), 7S (vicilin-like proteins) globulins, and 2S albumins. As expected, the 11S storage globulins were found in increased abundance, mainly in the dehulled seeds; in contrast, 7S globulins were more abundant in the hulls, and 2S albumins were ambiguously represented within the two cultivars evaluated.

The relative quantification of the data revealed that the second most abundant protein class (next to storage proteins) included oleosins as key proteins of the oil-body membranes. Metabolically important classes of proteins (e.g., proteins related to energy acquisition, nucleic acid metabolism, or protein synthesis) and proteins that are part of defence mechanisms or stress responses are more abundant in the hulls. The hulls of the hemp seed can thus be an important source of valuable proteins for use in food, medical, or biotechnological applications.

Although protein samples of two cultivars were evaluated, it became very clear that the cultivar is an important factor in the relative abundance of proteins, and the selection of a suitable cultivar will be important not only in terms of hemp cultivation as a field crop and in terms of seed-to-oil processing but also in the utilisation of the protein component of the seed.

## Figures and Tables

**Figure 1 plants-13-00111-f001:**
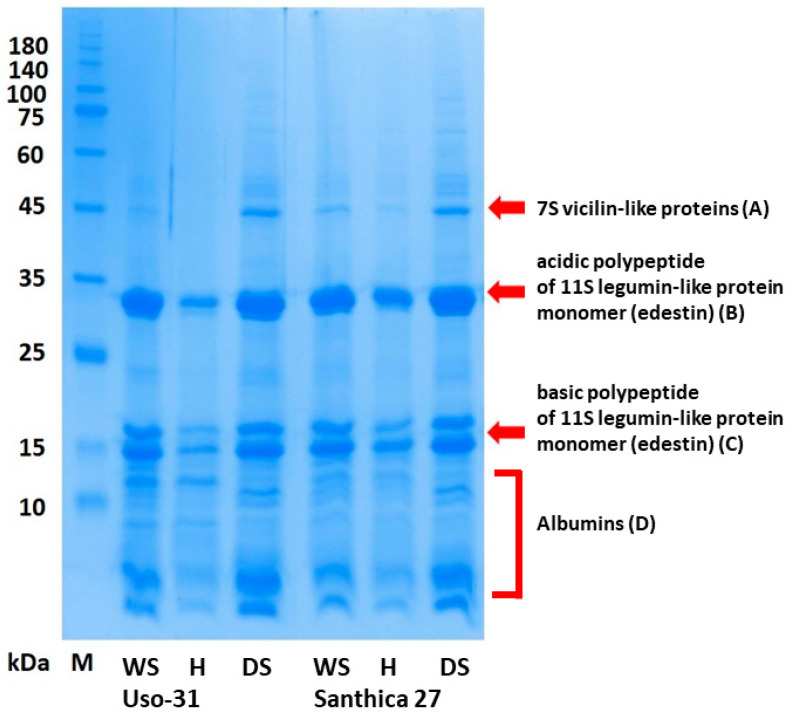
SDS-PAGE profiles of evaluated defatted products derived from seeds of two hemp cultivars under reducing conditions (WS—defatted flour from whole seeds; H—defatted flour from milled hulls; DS—defatted flour from dehulled hemp seeds, M—protein molecular weight standard ROTI®Mark Tricolor Xtra, Carl Roth GmbH + Co. KG, Karlsruhe, Germany).

**Figure 2 plants-13-00111-f002:**
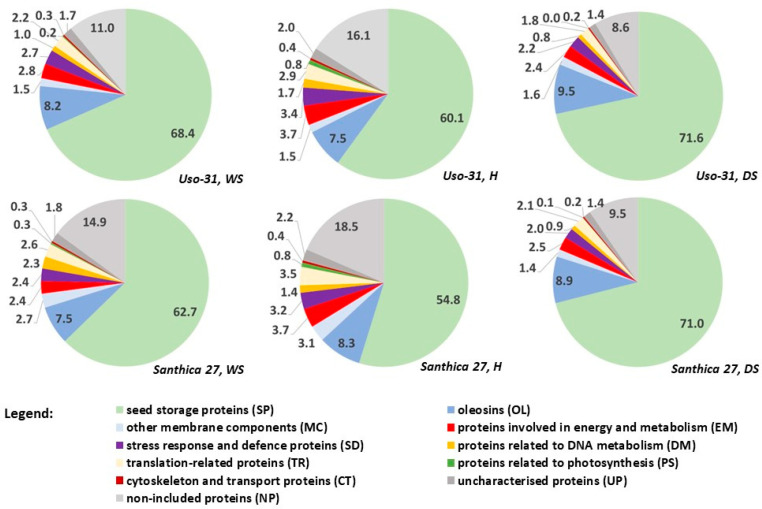
Pie charts of protein class sum proportions (in %) derived from relative abundance values in evaluated hempseed materials (WS—defatted flour from whole seeds; H—defatted flour from milled hulls; DS—defatted flour from dehulled hemp seeds).

**Table 1 plants-13-00111-t001:** Dry matter, protein, and fat contents in evaluated hemp seed samples (mean ± standard deviation).

Material	Cultivar	Variant	Dry Matter Content (% FM)	Protein Content (% DM)	Fat Content (% DM)
Original	Uso-31	whole seeds	92.01 ± 0.07 b	25.79 ± 1.68 e	29.40 ± 0.87 b
		hulls	91.57 ± 0.13 bc	13.40 ± 0.16 h	6.19 ± 0.19 d
		dehulled seeds	94.84 ± 0.07 a	35.35 ± 0.07 d	49.07 ± 0.41 a
	Santhica 27	whole seeds	91.39 ± 0.05 bc	27.74 ± 1.70 e	28.65 ± 0.63 b
		hulls	91.39 ± 0.86 bc	19.20 ± 0.62 g	15.43 ± 0.92 c
		dehulled seeds	93.87 ± 0.62 a	35.61 ± 0.31 cd	49.34 ± 0.79 a
Defatted	Uso-31	whole seeds	89.96 ± 0.17 d	37.85 ± 0.83 c	1.18 ± 0.26 e
		hulls	91.19 ± 0.09 bc	17.08 ± 0.18 g	0.05 ± 0.09 e
		dehulled seeds	91.38 ± 0.25 bc	71.74 ± 0.15 a	0.27 ± 0.15 e
	Santhica 27	whole seeds	89.76 ± 0.06 d	35.83 ± 0.89 cd	0.64 ± 0.13 e
		hulls	90.63 ± 0.20 cd	21.79 ± 0.21 f	0.23 ± 0.19 e
		dehulled seeds	90.09 ± 0.25 d	68.50 ± 0.44 b	0.04 ± 0.04 e

DM—dry matter; FM—fresh matter; Different letters in columns indicate the statistically significant difference at *p* < 0.05 (Tukey HSD test).

**Table 2 plants-13-00111-t002:** Description of evaluated proteins (information from UniProt KB database) included in MP88 collection.

Accession	Protein Name	Mol. Mass (Da)	Protein Families	Biological Process—Molecular Function—Cellular Component
**Seed storage proteins (SP)**				
A0A7J6GWL5	Cupin type-1 domain-containing protein	109,151	11S seed storage protein—globulins family	x—nutrient reservoir activity—x
A0A7J6DTA7	Cupin type-1 domain-containing protein	52,031	11S seed storage protein—globulins family	x—nutrient reservoir activity—x
A0A7J6E205	Cupin type-1 domain-containing protein	62,389	11S seed storage protein—globulins family	x—nutrient reservoir activity—membrane
A0A7J6H2R3	Cupin type-1 domain-containing protein	108,332		x—x—membrane
A0A7J6GLH5	Cupin type-1 domain-containing protein	108,083		x—x—membrane
A0A7J6HAT3	Cupin type-1 domain-containing protein	25,314		x—x—x
A0A7J6G321	Cupin type-1 domain-containing protein	55,360		x—x—x
A0A7J6H292	Bifunctional inhibitor/plant lipid transfer protein/seed storage helical domain-containing protein	16,752	2S seed storage albumins family; Plant LTP family	x—lipid binding; nutrient reservoir activity—x
A0A7J6DXD1	Bifunctional inhibitor/plant lipid transfer protein/seed storage helical domain-containing protein	17,558	2S seed storage albumins family; Plant LTP family	x—lipid binding; nutrient reservoir activity—x
**Oleosins (OL)**				
A0A7J6EJ89	Oleosin	15,410	Oleosin family	reproductive process; post-embryonic development—x—membrane; monolayer-surrounded lipid storage body
A0A7J6F0Y4	Oleosin	17,355	Oleosin family	reproductive process; post-embryonic development—x—membrane; monolayer-surrounded lipid storage body
A0A7J6H4F6	Oleosin	16,884	Oleosin family	reproductive process; post-embryonic development—x—membrane; monolayer-surrounded lipid storage body
**Other membrane components (MC)**				
A0A7J6I425	Verticillium wilt resistance-like protein	124,317	Receptor-like protein family	x—x—plasma membrane
A0A7J6F280	Peroxygenase	27,243	Caleosin family	x—x—membrane
A0A7J6G6U3	Uncharacterized protein	71,299		x—oxidoreductase activity—membrane
A0A7J6DNR1	Uncharacterized protein	64,236		x—x—membrane
A0A7J6DPR7	Uncharacterized protein	20,132		x—x—membrane
A0A7J6ENC9	Uncharacterized protein	19,908		x—transmembrane transporter activity—membrane
A0A7J6GSC3	Aquaporin TIP3-2	27,504	MIP/aquaporin (TC 1.A.8) family	x—channel activity—membrane
A0A7J6I7S9	Uncharacterized protein	41,884		x—x—membrane
**Proteins involved in energy and metabolism (EM)**				
A0A7J6G6Z3	Glyceraldehyde 3-phosphate dehydrogenase NAD(P) binding domain-containing protein	31,874	Glyceraldehyde-3-phosphate dehydrogenase family	x—NAD binding; oxidoreductase activity, acting on the aldehyde or oxo group of donors, NAD or NADP as acceptor —x
A0A7J6E4U9	Triose-phosphate isomerase	30,435	Triosephosphate isomerase family	glycolytic process—triose-phosphate isomerase activity—x
A0A7J6EFG0	Uncharacterized protein	46,262		x—oxidoreductase activity—x
A0A7J6EJG0	Protein disulfide-isomerase	56,660	Protein disulfide isomerase family	x—protein disulfide isomerase activity—endoplasmic reticulum lumen
A0A7J6E5J2	Fructose-bisphosphate aldolase	38,370	Class I fructose-bisphosphate aldolase family	glycolytic process—fructose-bisphosphate aldolase activity—x
A0A7J6EZ77	Malate dehydrogenase	36,570	LDH/MDH superfamily, MDH type 1 family	malate metabolic process; tricarboxylic acid cycle—L-malate dehydrogenase activity—x
A0A7J6HK40	Glyceraldehyde 3-phosphate dehydrogenase NAD(P) binding domain-containing protein	31,950	Glyceraldehyde-3-phosphate dehydrogenase family	x—NAD binding; oxidoreductase activity, acting on the aldehyde or oxo group of donors, NAD or NADP as acceptor—x
A0A7J6GRW8	Phosphopyruvate hydratase	46,384	Enolase family	glycolytic process—magnesium ion binding; phosphopyruvate hydratase activity—phosphopyruvate hydratase complex
A0A7J6EG53	Peptidyl-prolyl cis-trans isomerase	18,171	Cyclophilin-type PPIase family	protein folding; protein peptidyl-prolyl isomerization—peptidyl-prolyl cis-trans isomerase activity—x
A0A7J6FNP6	Malate synthase	63,098	Malate synthase family	glyoxylate cycle; tricarboxylic acid cycle—malate synthase activity—glyoxysome
A0A7J6E8J3	Malate dehydrogenase	36,478	LDH/MDH superfamily, MDH type 2 family	malate metabolic process; tricarboxylic acid cycle; malate metabolic process; tricarboxylic acid cycle—L-malate dehydrogenase activity—membrane
A0A7J6DST1	NADP-dependent oxidoreductase domain-containing protein	38,559	Aldo/keto reductase family	x—oxidoreductase activity—x
A0A7J6HKH5	Nucleoside-diphosphate kinase	16,290	NDK family	CTP biosynthetic process; GTP biosynthetic process; UTP biosynthetic process—nucleoside diphosphate kinase activity—x
A0A7J6HTX3	Tyrosinase copper-binding domain-containing protein	66,851	Tyrosinase family	pigment biosynthetic process—catechol oxidase activity; metal ion binding—x
A0A7J6HQA0	NADH-cytochrome b5 reductase	31,100	Flavoprotein pyridine nucleotide cytochrome reductase family	x—cytochrome-b5 reductase activity, acting on NAD(P)H—membrane
**Stress response and defence proteins (SD)**				
A0A7J6DVP5	rRNA N-glycosylase	28,760	Ribosome-inactivating protein family	defence response; negative regulation of translation—rRNA N-glycosylase activity; toxin activity—x
A0A7J6FXL4	Dehydrin	29,312	Plant dehydrin family	response to abscisic acid; response to cold; response to water deprivation—metal ion binding—x
A0A7J6I6U6	Uncharacterized protein	59,963	LEA type 4 family	x—x—x
A0A7J6G382	Uncharacterized protein	17,523	Small heat shock protein (HSP20) family	response to stress—x—x
A0A7J6HZ01	Annexin	36,065	Annexin family	response to stress—calcium ion binding; calcium-dependent phospholipid binding—x
A0A7J6I4E9	18 kDa seed maturation protein	15,710	LEA type 1 family	embryo development ending in seed dormancy—x—x
A0A7J6FL33	Late embryogenesis abundant protein D-29	30,417		x—x—x
A0A7J6GTA4	SHSP domain-containing protein	17,897	Small heat shock protein (HSP20) family	x—x—x
A0A7J6FPH5	Lactoylglutathione lyase	32,764	Glyoxalase I family	x—lactoylglutathione lyase activity; metal ion binding—x
A0A7J6DT98	Peroxiredoxin	24,133	Peroxiredoxin family, Prx6 subfamily	x—thioredoxin-dependent peroxiredoxin activity—x
A0A7J6GJC9	Dehydroascorbate reductase	23,719	GST superfamily, DHAR family	ascorbate glutathione cycle—glutathione dehydrogenase (ascorbate) activity; glutathione transferase activity—x
A0A7J6F3P9	Catalase	57,402	Catalase family	hydrogen peroxide catabolic process; response to oxidative stress—catalase activity; heme binding; metal ion binding—x
A0A7J6HTR6	Glutaredoxin domain-containing protein	14,424	Glutaredoxin family, CPYC subfamily	x—x—x
A0A7J6FRN5	Alcohol dehydrogenase	41,112	Zinc-containing alcohol dehydrogenase family	x—oxidoreductase activity; zinc ion binding—x
**Proteins related to DNA and RNA metabolism (DM)**				
A0A7J6EYT6	PPC domain-containing protein	30,708		x—minor groove of adenine-thymine-rich DNA binding—x
A0A7J6E9G4	Histone H4	11,409	Histone H4 family	x—DNA binding; protein heterodimerization activity; structural constituent of chromatin—nucleosome; nucleus
A0A7J6E6Z7	Histone H2A	15,148	Histone H2A family	x—DNA binding; protein heterodimerization activity; structural constituent of chromatin—nucleosome; nucleus
A0A7J6E9K3	Histone H2B	16,235	Histone H2B family	x—DNA binding; protein heterodimerization activity; structural constituent of chromatin —nucleosome; nucleus
A0A7J6HBN5	ATP-dependent RNA helicase	33,486	DEAD box helicase family	x—ATP binding; hydrolase activity; RNA binding; RNA helicase activity—x
**Translation-related proteins (TR)**				
A0A7J6HCW3	Elongation factor 1-alpha	49,259	TRAFAC class translation factor GTPase superfamily, Classic translation factor GTPase family, EF-Tu/EF-1A subfamily	x—GTP binding; GTPase activity; translation elongation factor activity—x
A0A7J6GTP8	60S acidic ribosomal protein P0	33,920	Universal ribosomal protein uL10 family	ribosome biogenesis—x—ribonucleoprotein complex; ribosome
A0A7J6HHK4	Translation elongation factor EF1B beta/delta subunit guanine nucleotide exchange domain-containing protein	25,124	EF-1-beta/EF-1-delta family	x—translation elongation factor activity—eukaryotic translation elongation factor 1 complex
A0A7J6F1D8	KH type-2 domain-containing protein	28,920	Universal ribosomal protein uS3 family	translation—RNA binding; structural constituent of ribosome—ribonucleoprotein complex; ribosome
A0A7J6HVI4	Ribosomal_L28e domain-containing protein	16,563	Eukaryotic ribosomal protein eL28 family	translation—structural constituent of ribosome—ribonucleoprotein complex; ribosome
A0A7J6E6C3	Ribosomal protein L7	30,458	Universal ribosomal protein uL30 family	maturation of LSU-rRNA from tricistronic rRNA transcript (SSU-rRNA, 5.8S rRNA, LSU-rRNA)—structural constituent of ribosome—cytosolic large ribosomal subunit
A0A7J6I4S9	Ribosomal protein L6 N-terminal domain-containing protein	25,942	Eukaryotic ribosomal protein eL6 family	translation—structural constituent of ribosome—ribonucleoprotein complex; ribosome
A0A7J6I083	60S acidic ribosomal protein P1	16,784	Eukaryotic ribosomal protein P1/P2 family	translational elongation—structural constituent of ribosome—ribonucleoprotein complex6; ribosome
A0A7J6F1J9	40S ribosomal protein S14	16,389	Universal ribosomal protein uS11 family	translation—structural constituent of ribosome—ribonucleoprotein complex; ribosome
A0A7J6GA02	Ribosomal_S7 domain-containing protein	22,343	Universal ribosomal protein uS7 family	translation—RNA binding; structural constituent of ribosome—small ribosomal subunit
A0A7J6G6F9	60S ribosomal protein L22-2; peroxidase	14,052	Eukaryotic ribosomal protein eL22 family; Peroxidase family	translation; hydrogen peroxide catabolic process; response to oxidative stress—structural constituent of ribosome; heme binding; lactoperoxidase activity—ribonucleoprotein complex; ribosome; extracellular region
A0A7J6I796	60S ribosomal protein L12	17,790	Universal ribosomal protein uL11 family	translation—structural constituent of ribosome—ribonucleoprotein complex; ribosome
A0A7J6HT66	40S ribosomal protein S9-2	23,142	Universal ribosomal protein uS4 family	translation—rRNA binding; structural constituent of ribosome—small ribosomal subunit
A0A7J6ENY3	RRM domain-containing protein	11,250		x—nucleic acid binding—x
A0A7J6FAW9	40S ribosomal protein S18	17,635	Universal ribosomal protein uS13 family	translation—rRNA binding; structural constituent of ribosome—ribonucleoprotein complex; ribosome
A0A7J6I9M7	Ribosomal_L18e/L15P domain-containing protein	20,840	Eukaryotic ribosomal protein eL18 family	translation—mRNA binding; structural constituent of ribosome—ribonucleoprotein complex; ribosome
A0A7J6FX59	KOW domain-containing protein	16,767	Universal ribosomal protein uL24 family	translation—RNA binding; structural constituent of ribosome—large ribosomal subunit
A0A7J6F744	Ribosomal_L14e domain-containing protein	15,331	Eukaryotic ribosomal protein eL14 family	translation—RNA binding; structural constituent of ribosome—ribosome
A0A7J6E9P9	60S ribosomal protein L27	15,794	Eukaryotic ribosomal protein eL27 family	translation—structural constituent of ribosome—ribonucleoprotein complex; ribosome
A0A7J6H424	50S ribosomal protein L23, chloroplastic	17,400	Universal ribosomal protein uL23 family; DHBP synthase family	translation; riboflavin biosynthetic process—mRNA binding; rRNA binding; structural constituent of ribosome; 3,4-dihydroxy-2-butanone-4-phosphate synthase activity; GTP binding—ribonucleoprotein complex; ribosome
A0A7J6EFK0	40S ribosomal protein S17	16,185	Eukaryotic ribosomal protein eS17 family	translation—structural constituent of ribosome—ribonucleoprotein complex; ribosome
A0A7J6G5C7	60S ribosomal protein L23	14,997	Universal ribosomal protein uL14 family	translation—structural constituent of ribosome—ribonucleoprotein complex; ribosome
A0A7J6GWW0	40S ribosomal protein S26	14,763	Eukaryotic ribosomal protein eS26 family	translation—structural constituent of ribosome; nucleotidyltransferase activity—ribonucleoprotein complex; ribosome
A0A7J6E4Y6	60S acidic ribosomal protein P3	11,903	Eukaryotic ribosomal protein P1/P2 family	translational elongation—structural constituent of ribosome—ribonucleoprotein complex; ribosome
**Proteins related to photosynthesis (PS)**				
A0A7J6GUI2	Chlorophyll a-b binding protein, chloroplastic	28,276	Light-harvesting chlorophyll a/b-binding (LHC) protein family	photosynthesis; light harvesting—chlorophyll binding—chloroplast thylakoid membrane; photosystem I; photosystem II
A0A7J6GSM8	Chlorophyll a-b binding protein, chloroplastic	28,429	Light-harvesting chlorophyll a/b-binding (LHC) protein family	photosynthesis, light harvesting—chlorophyll binding—chloroplast thylakoid membrane; photosystem I; photosystem II
**Cytoskeleton and transport proteins (CT)**				
A0A7J6I6X2	MD-2-related lipid-recognition domain-containing protein	16,057		intracellular sterol transport—sterol binding—x
A0A7J6I828	Actin	41,726	Actin family	x—ATP binding—cytoplasm; cytoskeleton
A0A0M5M1Z3	ATP synthase subunit alpha	55,324	ATPase alpha/beta chains family	actin filament bundle assembly; actin filament capping—actin filament binding; ATP binding; proton-transporting ATP synthase activity, rotational mechanism—mitochondrial inner membrane; proton-transporting ATP synthase complex, catalytic core F(1)
**Uncharacterised proteins (UP)**				
A0A7J6FR97	Uncharacterized protein	35,774		x—x—x
A0A7J6FP03	Uncharacterized protein	16,800		x—x—x
A0A7J6I334	Uncharacterized protein	15,922		x—x—x
A0A7J6GUW7	Uncharacterized protein	13,786		x—x—x
A0A7J6HWC7	Uncharacterized protein	17,135		x—x—x

x—information in UniProt KB database is not available.

**Table 3 plants-13-00111-t003:** Relative abundance of the MP88 collection proteins (in ppm) in defatted hemp seed products.

Accession	Protein Name	Uso-31			Santhica 27		
		Whole Seed	Hulls	Dehulled Seeds	Whole Seed	Hulls	Dehulled Seeds
A0A7J6GWL5	Cupin type-1 domain-containing protein	292,953 abc	249,256 cd	301,796 ab	265,551 bcd	225,575 d	316,781 a
A0A7J6DTA7	Cupin type-1 domain-containing protein	336,178 ab	290,488 bc	358,906 a	319,042 abc	278,002 c	350,150 a
A0A7J6E205	Cupin type-1 domain-containing protein	2991 b	734 c	5698 a	1767 bc	1546 bc	2397 b
A0A7J6H2R3	Cupin type-1 domain-containing protein	6299 ab	7324 a	6399 ab	5745 b	6426 ab	6214 ab
A0A7J6GLH5	Cupin type-1 domain-containing protein	3410 a	3532 a	3127 a	2708 a	2851 a	2780 a
A0A7J6HAT3	Cupin type-1 domain-containing protein	5750 bc	7593 a	5372 c	5890 bc	6958 ab	5461 c
A0A7J6G321	Cupin type-1 domain-containing protein	16,117 b	18,916 a	15,978 b	10,262 d	13,227 c	10,338 d
A0A7J6H292	Bifunctional inhibitor/plant lipid transfer protein/seed storage helical domain-containing protein	19,306 a	21,984 a	17,488 a	15,188 a	12,731 a	15,075 a
A0A7J6DXD1	Bifunctional inhibitor/plant lipid transfer protein/seed storage helical domain-containing protein	1052 ab	817 bc	1297 a	929 bc	681 c	909 bc
A0A7J6EJ89	Oleosin	16,729 bc	7800 d	26,828 a	12,215 cd	10,393 cd	21,992 ab
A0A7J6F0Y4	Oleosin	40,499 a	41,904 a	38,200 a	38,126 a	44,509 a	38,168 a
A0A7J6H4F6	Oleosin	25,020 a	25,250 a	29,762 a	24,470 a	28,268 a	28,554 a
A0A7J6I425	Verticillium wilt resistance-like protein	988 a	1458 a	1525 a	1547 a	1874 a	1371 a
A0A7J6F280	Peroxygenase	4675 ab	5415 a	4283 b	4015 b	4225 b	4176 b
A0A7J6G6U3	Uncharacterized protein	1205 a	1115 ab	1164 a	749 c	898 bc	721 c
A0A7J6DNR1	Uncharacterized protein	145 b	81 b	52 b	13,567 a	16,616 a	95 b
A0A7J6DPR7	Uncharacterized protein	1549 ab	1264 bc	1966 a	703 d	775 d	968 cd
A0A7J6ENC9	Uncharacterized protein	1498 c	2198 ab	1177 c	1561 bc	2410 a	1395 c
A0A7J6GSC3	Aquaporin TIP3-2	2236 a	2728 a	2585 a	3037 a	3023 a	2988 a
A0A7J6I7S9	Uncharacterized protein	2546 abc	824 d	3336 a	1959 bc	1464 cd	2655 ab
A0A7J6G6Z3	Glyceraldehyde 3-phosphate dehydrogenase NAD(P) binding domain-containing protein	3130 bc	4596 a	2123 c	3207 bc	4012 ab	2765 c
A0A7J6E4U9	Triose-phosphate isomerase	2120 a	2489 a	1862 a	1912 a	2233 a	2080 a
A0A7J6EFG0	Uncharacterized protein	2577 b	3347 a	2398 b	1993 cd	2455 b	1774 d
A0A7J6EJG0	Protein disulfide-isomerase	948 bc	1218 ab	823 c	1315 ab	1597 a	1239 ab
A0A7J6E5J2	Fructose-bisphosphate aldolase	1637 cd	2756 b	1158 d	2346 b	3594 a	1745 c
A0A7J6EZ77	Malate dehydrogenase	999 bc	1648 a	732 c	1154 b	1552 a	836 bc
A0A7J6HK40	Glyceraldehyde 3-phosphate dehydrogenase NAD(P) binding domain-containing protein	926 ab	1276 a	684 b	898 ab	1277 a	702 b
A0A7J6GRW8	Phosphopyruvate hydratase	701 bc	1043 a	638 c	807 b	1142 a	560 c
A0A7J6EG53	Peptidyl-prolyl cis-trans isomerase	4223 b	6554 a	3436 b	3861 b	6019 a	3035 b
A0A7J6FNP6	Malate synthase	816 bc	1225 a	591 d	878 b	1284 a	667 cd
A0A7J6E8J3	Malate dehydrogenase	995 bc	1289 ab	730 c	1129 bc	1792 a	939 bc
A0A7J6DST1	NADP-dependent oxidoreductase domain-containing protein	7132 a	5649 ab	7532 a	1840 b	5212 ab	7102 a
A0A7J6HKH5	Nucleoside-diphosphate kinase	1096 b	1555 a	539 c	1050 b	1602 a	802 bc
A0A7J6HTX3	Tyrosinase copper-binding domain-containing protein	266 d	1265 b	1 e	611 c	1660 a	10 e
A0A7J6HQA0	NADH-cytochrome b5 reductase	959 bc	1153 ab	340 d	1244 ab	1389 a	607 cd
A0A7J6DVP5	rRNA N-glycosylase	485 bc	2044 a	60 d	331 c	584 b	71 d
A0A7J6FXL4	Dehydrin	1169 a	983 a	1413 a	1457 a	1723 a	1265 a
A0A7J6I6U6	Uncharacterized protein	877 b	1277 a	711 bc	840 bc	1297 a	634 c
A0A7J6G382	Uncharacterized protein	5578 b	7698 a	4057 c	2919 de	3976 cd	2535 e
A0A7J6HZ01	Annexin	1282 bc	1481 b	1040 cd	1193 cd	1791 a	961 d
A0A7J6I4E9	18 kDa seed maturation protein	3226 abc	3796 a	2444 bc	3612 a	3344 ab	2104 c
A0A7J6FL33	Late embryogenesis abundant protein D-29	2567 b	2793 b	1986 b	2815 b	3770 a	2076 b
A0A7J6GTA4	SHSP domain-containing protein	1359 b	2012 a	1083 c	693 de	929 cd	536 e
A0A7J6FPH5	Lactoylglutathione lyase	2462 ab	2739 a	1947 c	2097 bc	2472 ab	1897 c
A0A7J6DT98	Peroxiredoxin	3473 bc	4054 b	2623 c	4150 b	6328 a	3130 bc
A0A7J6GJC9	Dehydroascorbate reductase	868 a	1052 a	703 ab	480 b	733 ab	724 ab
A0A7J6F3P9	Catalase	600 bc	1393 a	323 d	777 b	1421 a	399 cd
A0A7J6HTR6	Glutaredoxin domain-containing protein	2690 ab	1809 b	3306 a	2262 ab	2009 ab	2969 ab
A0A7J6FRN5	Alcohol dehydrogenase	843 b	1160 a	620 cd	690 bc	1129 a	452 d
A0A7J6EYT6	PPC domain-containing protein	1681 b	1234 b	2057 b	14,310 a	1344 b	2149 b
A0A7J6E9G4	Histone H4	4793 cd	9218 a	3183 e	5008 c	7431 b	3664 de
A0A7J6E6Z7	Histone H2A	1523 cd	3063 a	829 d	1950 bc	2576 ab	1207 cd
A0A7J6E9K3	Histone H2B	1251 bc	1921 a	1166 bc	1116 bc	1558 ab	858 c
A0A7J6HBN5	ATP-dependent RNA helicase	686 d	1147 b	464 e	821 c	1293 a	655 d
A0A7J6HCW3	Elongation factor 1-alpha	2695 bc	3321 ab	2066 c	2668 bc	3676 a	2345 c
A0A7J6GTP8	60S acidic ribosomal protein P0	882 cd	1325 b	749 d	1117 bc	1698 a	825 cd
A0A7J6HHK4	Translation elongation factor EF1B beta/delta subunit guanine nucleotide exchange domain-containing protein	811 abc	946 ab	593 c	932 ab	1065 a	749 bc
A0A7J6F1D8	KH type-2 domain-containing protein	696 de	899 bc	515 e	973 b	1239 a	727 cd
A0A7J6HVI4	Ribosomal_L28e domain-containing protein	1077 bc	1341 ab	728 c	1301 abc	1864 a	1283 abc
A0A7J6E6C3	Ribosomal protein L7	691 cd	1031 ab	530 d	854 bc	1163 a	618 d
A0A7J6I4S9	Ribosomal protein L6 N-terminal domain-containing protein	620 cd	815 b	562 d	759 bc	1061 a	636 bcd
A0A7J6I083	60S acidic ribosomal protein P1	753 d	977 bc	803 cd	1045 ab	1204 a	854 bcd
A0A7J6F1J9	40S ribosomal protein S14	1583 bc	1859 b	1190 c	1582 bc	2413 a	1300 c
A0A7J6GA02	Ribosomal_S7 domain-containing protein	804 ab	893 ab	593 b	703 b	1360 a	683 b
A0A7J6G6F9	60S ribosomal protein L22-2; peroxidase	1520 bc	1897 ab	1152 c	1664 bc	2307 a	1241 c
A0A7J6I796	60S ribosomal protein L12	869 bc	1133 ab	685 c	948 bc	1279 a	785 c
A0A7J6HT66	40S ribosomal protein S9-2	915 b	1336 a	544 c	961 b	1286 a	728 bc
A0A7J6ENY3	RRM domain-containing protein	397 b	1022 a	403 b	1068 a	1321 a	525 b
A0A7J6FAW9	40S ribosomal protein S18	733 b	1031 ab	673 b	1000 ab	1212 a	806 b
A0A7J6I9M7	Ribosomal_L18e/L15P domain-containing protein	886 bcd	1093 b	663 d	936 bc	1458 a	793 cd
A0A7J6FX59	KOW domain-containing protein	982 ab	955 ab	815 b	804 b	1453 a	953 ab
A0A7J6F744	Ribosomal_L14e domain-containing protein	1065 b	1480 a	683 c	1047 b	1470 a	929 bc
A0A7J6E9P9	60S ribosomal protein L27	835 b	1374 a	767 b	1211 a	1216 a	951 b
A0A7J6H424	50S ribosomal protein L23, chloroplastic	782 ab	767 ab	662 b	784 ab	1018 a	777 ab
A0A7J6EFK0	40S ribosomal protein S17	607 bc	793 b	545 c	726 bc	1017 a	589 bc
A0A7J6G5C7	60S ribosomal protein L23	651 b	603 bc	371 c	805 ab	1065 a	647 b
A0A7J6GWW0	40S ribosomal protein S26	880 bc	1124 ab	722 c	1100 ab	1460 a	826 bc
A0A7J6E4Y6	60S acidic ribosomal protein P3	540 a	570 a	588 a	759 a	1068 a	627 a
A0A7J6GUI2	Chlorophyll a-b binding protein, chloroplastic	434 bc	2459 a	54 c	864 b	2696 a	155 bc
A0A7J6GSM8	Chlorophyll a-b binding protein, chloroplastic	1449 c	6038 a	203 d	2560 b	5606 a	522 d
A0A7J6I6X2	MD-2-related lipid-recognition domain-containing protein	1140 a	1477 a	1082 a	1430 a	1696 a	1194 a
A0A7J6I828	Actin	973 a	1218 a	928 a	1019 a	1255 a	806 a
A0A0M5M1Z3	ATP synthase subunit alpha	549 bc	1081 a	377 d	629 b	1035 a	405 cd
A0A7J6FR97	Uncharacterized protein	1215 a	1071 a	1257 a	1047 a	1123 a	1174 a
A0A7J6FP03	Uncharacterized protein	12,415 ab	14,264 a	9720 b	13,312 ab	16,039 a	10,073 b
A0A7J6I334	Uncharacterized protein	1235 bc	1552 a	852 d	1149 cd	1479 ab	948 cd
A0A7J6GUW7	Uncharacterized protein	786 ab	856 ab	643 b	777 ab	1061 a	692 b
A0A7J6HWC7	Uncharacterized protein	1506 cd	2038 ab	1028 d	1570 bc	2260 a	1045 d

Different lowercase letters in the rows indicate a statistically significant difference at the *p* < 0.05 (Tukey HSD test) among the defatted hemp seed products of two cultivars (for each protein separately).

## Data Availability

The data presented in this study are available on request from the corresponding author.

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
