# Peer review of "Proteomic Profiles of Whole Seeds, Hulls, and Dehulled Seeds of Two Industrial Hemp (Cannabis sativa L.) Cultivars"

_plants, 2023, doi:10.3390/plants13010111_

Round 1

Reviewer 1 Report

Comments and Suggestions for Authors

The title "Quantitative Proteomic Profiles of Whole Seeds Hulls and Dehulled Seeds of Two Industrial Hemp (Cannabis sativa L.) Cultivars" accurately reflects the content and focus of the document. It centers on the quantitative proteomic analysis of different parts of the hemp seed, including the hulls and dehulled seeds, from two industrial hemp varieties.

In the abstract, when the authors refer to "two cultivars," do these have any registration or traceability? If so, please provide the ID of each.

The abstract mentions, "Seed storage proteins were found to be the most abundant protein class." Could you please indicate the percentage of this abundance?

The authors have described the most significant findings of their research well. However, two crucial aspects need to be included in this section. The first is the objective and motivation of the work. Why study the proteome of different parts of the seed? What is the implication of this in germination? The second aspect that needs to be added is the conclusion of the work. What understanding was gained from this proteomic analysis? These questions should be straightforward in the abstract.

The introduction mentions the species Cannabis indica. However, the authors should search in WFO and other databases, as many species are synonyms of Cannabis sativa L. In this case, it should be mentioned if it is a hybridization or what race or variety it is.

What is the THC concentration that industrial hemp should have? Please clarify.

What is the percentage of polyunsaturated fatty acids in the hemp seed? This is to compare with other seeds in nutritional aspects. And what are the present polyunsaturated fatty acids?

References supporting statements in lines 72 to 76 need to be included.

The introduction describes the seeds' nutritional properties and compares them with other crops. However, the proteomic part needs to be clarified in the introduction. The importance of the LC-MS/MS technique for its analysis needs to be mentioned, and the concrete objective of the research needs to be clarified. In the last paragraph, the authors set out three objectives, the second and third of which are not mentioned in the introduction. Therefore, it is necessary to address biological processes, molecular function, or cellular structure, and finally, how and what relationship there is in "comparing the effect of seed product type and cultivar on the relative abundance of significant proteins."

The authors have done an excellent job of identifying the proteins in the three selected tissues. They have identified 2833 proteins in hemp seed samples, with 88 proteins representing 81.5-91.4% of the total protein amount. This highlights different classes of proteins and their relative abundance in hemp seeds and hulls. The importance of these proteins in food, medical, and biotechnological applications is discussed. The significance of the cultivar in the relative abundance of proteins is emphasized.

The conclusions are relevant to the presented results.

The manuscript submitted by the authors is engaging and meets the journal's scope. The details to be refined are minor but essential to highlight the significance of the research. The manuscript's primary effort is centered on the results and discussions section. Therefore, the abstract and introduction require substantial improvements to identify the problem the authors wish to address and pose a straightforward research question. Based on these two aspects, I will recommend major reconsiderations to review the manuscript in this section again. With this in mind, I recommend the authors make minimal modifications in results, discussions, and conclusions to avoid generating new manuscript versions.

Author Response

Dear reviewer,
I am sending the file enclosed with answers to the comments and suggestions. Thank you for the notices that helped us improve the manuscript.

Best regards
Veronika Bartova

Reviewer 2 Report

Comments and Suggestions for Authors

The manuscript aims at characterize the proteomic profile of different samples of Cannabis sativa (2 varieties), namely whole seeds, hulls and dehulled seeds. The most important proteins were classified in different groups according to the the function and the relative abundance was calculated for each of them.

Generally speaking the paper was well written and experiments well designed. Investigations aimed at deepen the knowledge on the nutritional value of food and relative by products are always intriguing for the impact they could have in the field of sustainable economy, valorization and re-cycle of food by products, that could represent good sources of nutrients.

Here some comments for the authors:

1) Pag 16, line 414. For clarity purpose, please the authors specified that dry matter, protein and fat content were estimated both in original and defatted samples (whole seeds, hulls and dehulled seeds).

2) Pag 4, line 161. The section about the proteomic investigation should be better introduced. Please the authors specified which samples they used for protemic investigation (I suppose the tryptic digest of the whole samples) and how did they perform the quantification. Throught the text they always discuss the relative abundances of the proteins expressed in percentage, but in the table 3 they report the relative abundance in ppm. How did they convert this relative abundance in ppm? Or how did they quantify the proteins? Are the numbers reported in table 3 deriving from calculations based on peak areas? Please the authors clarify this crucial point because it is a bit confusing…In general “ppm” refers to a concentration with a specific unit of measure, for example mg/kg, but I think this is not the case.

3) Pag. 5 line 165. How did the authors estimated a relative abundance greater than 1000ppm?

4). Pag 5, table 2: please the authors details the meaning of x-x-x in the table.

5) Pag. 12 line 331. As stated by the authors, the reserve compounds, such as seed storage proteins, should be mainly located in the inner part of the seed. Therefore, how did they explain a 60% of seed storage proteins in the hulls? Did the authors have some information about the dehulling process?

6) Pag. 3 table 1. How did the authors explain the higher protein content observed in the dehulled samples with respect the whole seeds?

7) Considering that protein content of each sample was reported only as relative abundance, I would substitute the term “quantification” reported throughout the text with “semiquantitative analysis” because “quantification” is misleading with respect the investigation done. To quantify a compound the relative standard and curve calibration are generally required thus to express the calculated amount of the molecule in mg/kg.  

Author Response

(The authors gave the same response as above.)

Reviewer 3 Report

Comments and Suggestions for Authors

In this article, the authors compared proteins identified from whole seeds, hulls and dehulled seeds of two hemp cultivars using LC-MS/MS-based proteomics method. Among 2833 identified proteins, most of the 88 high-abundance proteins were confirmed and studied based on the literatures. Moreover, the protein profile of hulls were studied. According to the study, the authors claimed the cultivar selection is important for food industry. While the work describes a complete story, I have several questions and suggestions here.

1. The quantifiable data selection (page 5, line 165) is “in at least one of the three independent replicates for one of the three sample types”. This seems not reasonable for proteome analyses. Usually at least 60% valid data was required for further quantitative analysis, which means at least two of three replicates or three out of five. How the proteins look like if stricter criteria is applied?

2. The trypsin to protein ratio seems low compared to standard proteomics. Please specify the temperature and incubation time for digestion. 

3. In the part of seed storage proteins, three identified 11S proteins are similar to previous reported edestins as for its sequence. But what does that mean? The functions are similar as well? Or they are unstudied proteins?

4. Please clarify how the lower-case letters indicate the significance in Table 3. Are they referring to different groups or different level of significance?

Author Response

(The authors gave the same response as above.)

Round 2

Reviewer 1 Report

Comments and Suggestions for Authors

The authors have addressed each comment, provided a robust response, and made the requested modifications in the manuscript. I don't have any more comments to add to the article, so I recommend it should be accepted for publication.